# Evaluation of SARS-CoV-2 in the Vaginal Secretions of Women with COVID-19: A Prospective Study

**DOI:** 10.3390/jcm10122735

**Published:** 2021-06-21

**Authors:** Elad Barber, Michal Kovo, Sophia Leytes, Ron Sagiv, Eran Weiner, Orna Schwartz, Margarita Mashavi, Keren Holtzman, Jacob Bar, Anat Engel, Shimon Ginath

**Affiliations:** 1Wolfson Medical Center, Department of Obstetrics & Gynecology, E. Holon 5822012, Israel; michalkovo@gmail.com (M.K.); Leytes84@gmail.com (S.L.); sagivr@wmc.gov.il (R.S.); masolbarak@gmail.com (E.W.); jbar@wmc.gov.il (J.B.); ginath@gmail.com (S.G.); 2Sackler School of Medicine, Tel Aviv University, Tel Aviv 69978, Israel; Ornas@wmc.gov.il (O.S.); mashavi@wmc.gov.il (M.M.); kerenschweid@gmail.com (K.H.); AnatEngel@wmc.gov.il (A.E.); 3Wolfson Medical Center, Microbiology Laboratory, E. Holon 5822012, Israel; 4Wolfson Medical Center, Department of Internal Medicine “D,” E. Holon 5822012, Israel; 5Wolfson Medical Center, Department of Medical Administration, E. Holon 5822012, Israel

**Keywords:** COVID-19, SARS-CoV-2, vaginal secretions

## Abstract

Objective: We aimed to investigate the likelihood of vaginal colonization with Severe acute respiratory syndrome coronavirus 2 (SARS-CoV-2) in pregnant and non-pregnant women with Coronavrus Disease 2019 (COVID-19). Materials and Methods: Vaginal swabs were taken from women diagnosed with mild to moderately acute SARS-CoV-2 infection, at Wolfson Medical Center, Israel, from March 2020 through October 2020. COVID-19 was diagnosed by real-time polymerase chain reaction (RT-PCR) performed on nasopharyngeal swabs. Vaginal swabs were tested for the presence of SARS-CoV-2 by reverse transcription polymerase chain reaction (RT-PCR). Results: In total, 51 women diagnosed with COVID-19 were included in the study. Of the 51 women with COVID-19 enrolled in this study, 16 (31.4%) were pregnant at enrollment and 35 (68.6%) were non-pregnant. Mean age was 43.5 ± 15.3 years (range 21–74 years). Compared to the non-pregnant group, the pregnant group was characterized by a higher white blood cell and absolute neutrophil count (*p* = 0.02 and *p* = 0.027, respectively). The non-pregnant patients were more likely to have chronic diseases (*p* = 0.035) and to be hospitalized (*p* < 0.001). Only one patient (1.9%) aged 60 years tested positive for SARS-CoV-2 in vaginal secretions. Mean gestational age at the diagnosis of COVID-19 of the pregnant group was 32.3 ± 7.8 weeks. Thirteen patients delivered during the study period; all delivered at term without obstetric complications and all neonates were healthy. Conclusions: Detection of SARS-CoV-2 in the vaginal secretions of patients diagnosed with COVID-19 is rare. Vaginal colonization may occur during the viremia phase of the disease, although infectivity from vaginal colonization needs to be proven.

## 1. Introduction

The coronavirus pandemic outbreak of 2019 (COVID-19) is a novel disease that has not been previously described in humans. Studies have made progress regarding COVID–19 disease mechanisms, but many aspects have yet to be clarified and continue to challenge the scientific community, mainly the mechanisms that are responsible for the infection and the deterioration of the disease [1]. Reaching the goal of effective treatment and vaccine regimens requires a greater understanding of the pathomechanisms of COVID–19 and its associated complications [1].

The pathogen was identified as a new coronavirus, named severe acute respiratory syndrome coronavirus 2 (SARS-CoV-2). Since December 2019, the scientific community has been on a race to study the virus, including understanding the pathogenesis and transmission of the disease and defining the risk factors [2]. Importantly, data are lacking regarding vertical transmission of the virus and whether it has deleterious neonatal effects [3,4].

It belongs to a group of human coronaviruses, such as SARS-CoV and the Middle East respiratory syndrome coronavirus (MERS-CoV), which were responsible for outbreaks that occurred in 2003 and 2012, respectively [5]. Since it was first reported in Wuhan, Hubei Province, China [6], COVID-19 has affected the entire world in many aspects, including health, economics, education, and culture. 

Studies have revealed that the SARS-CoV-2 virus is transmitted from human to human by respiratory droplets and direct contact through mucosal surfaces. There is preferential tropism to human airway epithelial cells through the cellular receptor angiotensin converting enzyme 2 (ACE2) [7,8]. Although the lungs are the primary organ involved, the virus has also been detected in the blood, urine, tears, and the gastrointestinal (GI) tract. The virus could not be detected in the semen of men acutely infected with SARS-CoV-2 or recovered from COVID-19, although semen parameters were affected. Consequently, orchitis may be a complication following COVID-19 [9,10].

There are inconsistent reports concerning the presence of SARS-CoV-2 in the female genital tract [11,12]. Vivanti et al. [13] were the first to describe the presence of SARS-CoV-2 in the vagina of a woman following a cesarean delivery, while Schwartz et al. [14] reported the presence of the virus in the vaginal secretions of 2 out of 35 infected patients. Other studies, mainly case reports and small cohorts, failed to demonstrate the presence of SARS-CoV-2 in the vaginal secretions of pregnant and non-pregnant women [7,8,11,15,16,17].

We aimed to investigate the presence of SARS-CoV-2 in vaginal secretions in a large cohort of pregnant and non-pregnant women with COVID-19.

## 2. Materials and Methods

This was a prospective study. The study was approved by the local institutional ethical committee (number 0110-20-WOMC).

Women who had been diagnosed with acute SARS-CoV-2 infection were recruited at a single tertiary university-affiliated hospital, E. Wolfson Medical Center, Israel, from March 2020 through October 2020. All participants gave oral informed consent, which was witnessed by two medical staff members who signed the consent form. COVID-19 was diagnosed by the detection of SARS-CoV-2 RNA by real-time polymerase chain reaction (RT-PCR) obtained from nasopharyngeal swabs [18].

We included pregnant and non-pregnant women who had mild to moderately severe disease. All patients have had blood tests including white blood cell count and CRP as part of the standard work-up. Hospitalized patients have had chest X-rays done upon admission. Mild disease was defined when patients presented with normal blood oxygen levels and with no abnormal findings on chest X-ray. Moderate disease was defined when patients had blood oxygen levels below 95% or abnormal findings on chest X-ray. Lastly, severe disease was defined as abnormal findings on chest X-ray and the need for oxygen of at least 6 L per hour. 

Excluded from the study were patients under 18 years old who could not give legal consent to participate, severely ill patients not able to give informed consent, and patients who were unwilling to participate. 

Medical history, including background illnesses as well as obstetric and surgical history, was obtained by history taking from the patients as well as from their medical charts.

Close contact was made with all pregnant women during the study period (March 2020 through October 2020), which included obtaining updates on pregnancy outcomes, either by revisiting their medical charts or by phone calls made by the researcher E.B. 

### Sample Collection and Laboratory Analysis

Prior to vaginal sampling, the patients were asked to thoroughly clean the perineum with a septal scrub. Vaginal sampling was performed by a physician or by the patient under direct visualization. Vaginal swabs were obtained following US Center for Disease Control and Prevention (CDC) guidelines [19]. Vaginal swabs were inserted 2–3 cm into the vagina and rotated for several seconds [20]. After obtaining the samples, they were transferred immediately to the microbiology laboratory in a designated transport medium (viral transport medium, Biological Industries Israel^®^). RT-PCR was performed using accepted protocols [21]. The time interval from diagnosis of COVID-19 to vaginal swab collection was calculated in days. In case of positive vaginal sample results, we aimed to perform an additional sample to minimize the risk of a false-positive result.

RT-PCR testing for SARS-CoV-2 in our laboratory was performed utilizing one of the following two methods: the DiaSorin Molecular Simplexa™ COVID-19 Direct real-time RT-PCR assay (LIAISON^®^ MDX instrument, CYPRESS, CA, USA), which analyzes S and Orf1b genes of SARS-CoV-2, and GeneXpert^®^ Xpress SARS-CoV-2 (powered by Cepheid innovation, SUNNYVALE, CA, USA), which analyzes E and N2 genes of SARS-CoV-2. 

Positive results for both methods were determined according to known values [22,23]. Specifically, a positive result for genes E and N2 was determined with values of Ct < 45 [23]. The method used for each sample was chosen for convenience**.** In case of a positive result utilizing one of the methods, an additional vaginal sample was taken and a separate test was done utilizing the other method to strengthen the validity of the results. 

## 3. Results

Fifty-one women diagnosed with COVID-19 were included in the study; of them, 16 (31.3%) were pregnant. Out of the 51 patients, 45 (88%) were hospitalized and the rest were discharged from the hospital after a thorough clinical and laboratory work-up.

Table 1 presents the comparison between pregnant and non-pregnant patients. The non-pregnant group was characterized by a higher mean age (50.0 ± 13.7 vs. 29.5 ± 7.3 years, *p* < 0.001) and a higher rate of chronic diseases (51.4% vs. 18.7%, *p* = 0.035). With regard to laboratory tests on admission, the pregnant group was characterized by higher levels of white blood cell count (WBC; 10^3^/μL) and absolute neutrophil count (10^3^/μL; 7.6 ± 1.6 vs. 5.8 ± 2.8, *p* = 0.02, and 5.3 ± 1.6 vs. 3.9 ± 2.2, *p* = 0.027, respectively). The non-pregnant patients were more likely to be hospitalized for further treatment and observation (*p* < 0.001). X-ray findings were similar between the groups.

Table 2 presents the characteristics of the 16 pregnant women with COVID-19. The mean gestational age at diagnosis was 32.3 ± 7.8 weeks. Five patients were infected during the second trimester (15 to 28 weeks) and 11 in their third trimester (30 to 41 weeks). Thirteen patients (81.2%) delivered during the study period; among them, 7 (53.8%) delivered vaginally. Indications for cesarean delivery included cephalo-pelvic disproportion, non-reassuring fetal heart rate, and scheduled repeat cesarean delivery. All neonates were tested for SARS-CoV-2 during the first 24 h after birth. All were negative for the test and none required any special treatment.

Only one patient (1.9%) had positive results for SARS-CoV-2 in two separate samples of vaginal secretions. The patient was 60 years old with a significant medical history of chronic diseases: systemic lupus erythematosus, antiphospholipid syndrome, and hypertension. In her past, she underwent total abdominal hysterectomy due to menorrhagia and uterine leiomyomas. Her regular medications included methotrexate, warfarin, aspirin, levothyroxine, belimumab, and hydroxychloroquine. She was diagnosed with SARS-CoV-2 following complaints of the symptoms of dry cough and weakness and was hospitalized for supportive care. At admission and during her hospitalization, she was hemodynamically stable, without worsening of her symptoms. She was mobile, without other complaints. Repeat physical examinations were without additional pathological findings and she was discharged 4 days following her admission in good general condition. The first vaginal secretion sampling for SARS-CoV-2 was performed on the same day she was admitted and diagnosis of COVID-19 was made. Following the positive result of SARS-CoV-2 in vaginal secretions with the genes S and Orf1b (Figure 1B), we took an additional separate vaginal sample on the same day utilizing a different analyzing method that was positive as well—this time for the gene N2 (Figure 1C). Each separate test utilized a different method for the detection of the virus (Figure 1). Out of four genes, three were positive (N2, S, and Orf1b). All three genes were found to be positive but in very low levels compared to the results obtained from the nasopharyngeal sample.

## 4. Discussion

In the current study, the presence of the SARS-CoV-2 virus in the vaginal secretions of women diagnosed with COVID-19 was tested. The presence of SARS-CoV-2 was detected in only one post-menopausal patient of 51 women. Notably, the viral load was just at the limit of detection for the assay, much lower than the results obtained from the nasopharyngeal swab collected from the same woman.

Detection of SARS-CoV-2 in vaginal secretions might have numerous adverse effects including alterations in regular changes in endometrial tissue and embryo development [21]. Following the first report of the detection of SARS-CoV-2 in the vaginal secretions of a pregnant woman with COVID-19 [13], several other studies have failed to detect the virus in the vaginal secretions of women with COVID-19 [7,8,15]. Qiu et al. [7] did not detect the SARS-CoV-2 virus in the vaginal secretions of 10 patients, aged 52 to 80 years, who had been diagnosed with severe COVID-19. Notably, the time elapsed from diagnosis to sampling was 28.1 ± 6.1 days. Similar findings were described in 12 pregnant patients with mild symptoms of COVID-19 [8]. In concordance with the current study, a recent study by Schwartz et al. [14] succeeded in detecting SARS-CoV-2 in the vaginal secretions of 2 out of 35 women aged 21 and 93 years. The investigators used different but similar vaginal RT-PCR swabs compared to our study.

There are several possible explanations for the detection of SARS-CoV-2 in the vaginal secretions of only a few infected women with a rate of 1.9% in the current study and up to 5.7% in other reports [14]. First, the virus indeed could disseminate from the epithelial barrier of the lungs to other organs, including the central nervous system (CNS), through the vascular system [24,25]. Detection of SARS-CoV-2 RNA in the plasma has already been reported [26,27]. However, the low incidence of viremia (1–37%) in patients with COVID-19 could explain the low rate of its detection in vaginal secretions [28,29,30,31,32].

Second, the positive results could be due to “fecal contamination” or “ascending infection” of the virus from the GI tract [33]. However, it has been shown that the expression of the virus in the feces occurs relatively late in the disease course [34]. Notably, in the current study, vaginal sampling was performed near the time of diagnosis of COVID-19. Moreover, the patient who was found to have positive results in the vaginal secretions had her vaginal swabs obtained when she had a positive NP test as well, indicating high levels of the virus. Additionally, in order to reduce the possibility of contamination, sampling was performed after perineum cleaning had been done. Third, the results could be falsely positive [32,35]. We minimized this option by performing a second separate RT-PCR analysis of different genes, establishing the positive results we obtained.

The study is unique in several aspects. First, to our knowledge, it is the largest prospective study to investigate the presence of SARS-CoV-2 in the vaginal secretions of pregnant and non-pregnant women infected with SARS-CoV-2. By doing so, we looked for possible clinical characteristics of infected women that could distinguish the patients with a possible vaginal contamination with the SARS-CoV-2 virus. Second, we present additional data on pregnant women with COVID-19 during pregnancy, who have already delivered. Vertical transmission was ruled out by SARS-CoV-2 negative tests of the newborns. Third, samples were obtained close to the diagnosis of COVID-19.

Our study is not without limitations. First, simultaneous sampling of blood, urine, and feces was not performed. This could have contributed to our knowledge and established the speculation that the viremia stage is an important phase. Second, the sample of the study is too small to conclude a possible vertical transmission during pregnancy or possible effects of the COVID-19 during the first trimester.

In conclusion, detection of SARS-CoV-2 in vaginal secretions during COVID-19 infection is not common. In the current study, we detected the virus in one patient (1.9%). Possibly, detection of the virus is feasible only in the viremia stage of the disease. Larger studies are needed to establish the infectivity and risks that might be associated with vaginal colonization with SARS-CoV-2.

## Figures and Tables

**Figure 1 jcm-10-02735-f001:**
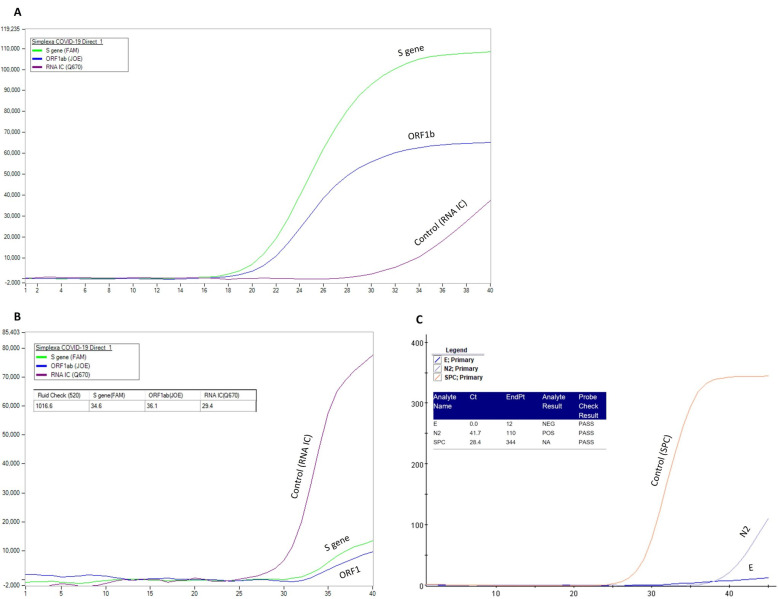
Real-time polymerase chain reaction results of the patient with positive SARS-CoV-2 in the vaginal secretions. (**A**) Results from nasopharyngeal swab utilizing the DiaSorin Molecular Simplexa™ COVID-19 Direct real-time RT-PCR assay, testing for genes S and Orf1b. (**B**) Results from the first positive vaginal swab utilizing the DiaSorin Molecular Simplexa™ COVID-19 Direct real-time RT-PCR assay, testing for genes S and Orf1b. (**C**) Results from the repeat vaginal swab utilizing GeneXpert^®^ Xpress SARS-CoV-2, testing for genes E and N2. Cycle threshold (Ct) levels are inversely proportional to the amount of mRNA of the targeted antigen. The Ct of the S gene, ORF1, and N2 in the vaginal samples was 34.6, 36.1, and 41.7, respectively (**B**,**C**). The *X* and *Y* axes represent the amount of amplified RNA and the number of cycles, respectively; the earlier the signal is detected, the lower the number of cycles and the higher the viral load. The orange line (**C**), sample processing control (SPC), and the purple line (**B**), RNA internal control (RNA IC), represent the positive controls. A positive result for the genes E and N2 is determined with values of Ct < 45 [22]. Positive results for both methods were determined according to known values [22,23]. Cycle threshold (Ct) levels are inversely proportional to the amount of mRNA of the targeted antigen. Ct of the S gene, ORF1 and N2 in the vaginal samples were 34.6, 36.1 and 41.7, respectively (**B**,**C**).

**Table 1 jcm-10-02735-t001:** Demographic and clinical characteristics of the pregnant and non-pregnant groups of women infected with Severe acute respiratory syndrome coronavirus 2 (SARS-CoV-2).

	Pregnant (*n* = 16)	Non-Pregnant (*n* = 35)	*p*-Value
Age (years) (Range 23–70)	29.5 ± 7.3	50.0 ± 13.7	<0.001
Gravity	2.8 ± 1.8	3.5 ± 2.9	0.379
Parity	1.5 ± 1.4	2.7 ± 1.9	0.028
BMI (kg/m^2^) (Range 20–44)	25.3 ± 3.0	28.2 ± 6.9	0.114
Chronic diseases	* 3 (18.7)	^†^ 18 (51.4)	0.035
Time interval from diagnosis of COVID-19 infection to obtaining vaginal samples (days) (Range 0–12)	2.9 ± 4.3	4.2 ± 5.8	0.427
Hospitalization	10 (62.5)	35 (100)	<0.001
Positive SARS-CoV-2 in vaginal samples	0	1 (2.8)	>0.999
Positive X-ray findings	2 (12.5)	12 (34.2)	0.176
White blood cells (10^3^/μL) (Range 2.7–13.1)	7.6 ± 1.6	5.8 ± 2.8	0.020
Neutrophils (10^3^/μL) (Range 1.3–10.7)	5.3 ± 1.6	3.9 ± 2.2	0.027
Lymphocytes (10^3^/μL) (Range 0.9–3.5)	1.6 ± 0.6	1.9 ± 4.2	0.778
CRP (mg/dl) (Range 0.5–11.4)	2.9 ± 2.3	3.6 ± 3.0	0.412
Platelets (10^3^/μL)(Range 140–363)	198.7 ± 48.6	212.5 ± 64.9	0.452

Continuous variables are presented as mean ± SD and categorical variables as *n* (%); *asthma, anemia, fibromyalgia; ^†^ thyroid cancer, chronic bronchitis, hypertension, diabetes, obesity, hypothyroidism, systemic lupus erythematosus, antiphospholipid antibody syndrome, ischemic heart disease, past thrombotic events; CRP: C-reactive protein.

**Table 2 jcm-10-02735-t002:** Characteristics and obstetric outcomes of pregnant women infected with SARS-CoV-2 who delivered during the study period.

	*n* = 13
Gestational age at delivery (weeks) (Range 36.0–41.1)	38.5 ± 1.8
Mode of labor (VD)	7 (53.8)
Neonatal birth weight (gram) (Range 2482–3605)	3177.2 ± 487.4
Neonatal birth weight (percentile) (Range 29–83)	56.2 ± 25.1
Neonatal cord pH (Range 7.16–7.36)	7.22 ± 0.1
Positive nasopharyngeal COVID-19 in neonates	0 (0)

Continuous variables are presented as mean ± SD and categorical variables as *n* (%), VD, vaginal delivery.

## Data Availability

The data presented in this study are available on request from the corresponding author. The data are not publicly available due to privacy concerns.

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
