# Peer review of "Evaluation of SARS-CoV-2 in the Vaginal Secretions of Women with COVID-19: A Prospective Study"

_jcm, 2021, doi:10.3390/jcm10122735_

Round 1

Reviewer 1 Report

The manuscript is well written and although it gives the impression of short communication, the methodology used is correct and the conclusions drawn from the study are valid.
Below I enclose notes to the manuscript:

1. The introduction is well written, but I am asking the authors to expand the literature. Please add this paper (doi: 10.3390/pathogens9060493) to the first sentence, and this article to the second sentence (doi: 10.3390/pathogens9030231).
2. the aim of the work is clear and does not need to be improved.
3. results are presented clearly - above all an interesting clinical observation. I encourage the authors to continue their research in this issue.

Author Response

Reviewer 1- general comment:

            We wish to thank the reviewer by the remarks. We have made significant changes throughout the manuscript to accommodate.

comment 1

The introduction is well written, but I am asking the authors to expand the literature. Please add this paper (doi: 10.3390/pathogens9060493) to the first sentence, and this article to the second sentence (doi: 10.3390/pathogens9030231).

Response 1

We thank the reviewer for the remark. We have expanded the introduction with the suggested articles:

Lines 33-38: Studies have made progress regarding COVID–19 disease mechanisms, but many aspects have yet to be clarified and continue to challenge the scientific community, mainly- the mechanisms which are responsible for the infection and the deterioration of the disease. [1] Reaching the goal of effective treatment and vaccine regimens is related to a greater understanding of the pathomechanisms of COVID–19 and its associated complications. [1]

Lines 40-42: Since December 2019, the scientific community has been on a race to study the virus including understanding the pathogenesis and transmission of the disease and define the risk factors. [2]

Reviewer 1 comments 2 & 3

The aim of the work is clear and does not need to be improved.
Results are presented clearly - above all an interesting clinical observation. I encourage the authors to continue their research in this issue.

Response 2 & 3

We wish to thank the reviewer for the comments.

References:

  1. Słomka A, Kowalewski M, Żekanowska E (2020) Coronavirus Disease 2019 (COVID-19): A Short Review on Hematological Manifestations. Pathogens 9 (6). doi:10.3390/pathogens9060493
  2. Rabi FA, Al Zoubi MS, Kasasbeh GA, Salameh DM, Al-Nasser AD (2020) SARS-CoV-2 and Coronavirus Disease 2019: What We Know So Far. Pathogens 9 (3). doi:10.3390/pathogens9030231

Reviewer 2 Report

Barber et al. have looked a larger cohort of women with COVID-19 than has been studied before to see if they can find evidence of SARS-CoV-2 in vaginal secretions. Of the 51 women they were able to enroll, only one woman’s vaginal swab tested positive, but the Ct values from those tests indicate very low amounts of virus. The authors need to provide more evidence that these Ct values indicate positive results on these tests.

Line 2: change to “Evaluation of SARS-CoV-2 in Vaginal Secretion of Women with COVID-19: A Prospective Study” (make sure o in CoV is lower case here and in the abstract)

Line 13: delete “infected”

Line 16: delete “infection”; change “obtained from the nasopharynx” to “performed on nasopharyngeal swabs”

Line 18: reword to “Of the 51 women with COVID-19 enrolled in this study, 16 (31.4%) were pregnant at enrollment.” (the percentage should be rounded up to 31.4)

Line 19: delete “As compared to the non-pregnant group”

Line 20: delete “level of”

Line 21: change to “counts” for both uses of count in this line

Line 25: change to “Thirteen patients delivered during the study period; all delivered at term without obstetric complications and all neonates were healthy”

Line 27: change “probably” to “may”

Line 28: change “disease. Yet,” to “disease, although”

Line 37: delete “ever”. Move the reference after “China” since it is about the first report, not about worldwide effects.

Line 40: either remove the hyphens in human-to-human or remove “from”

Line 42: either remove the dash or replace with a comma.

Line 43: change “major” to “primary”

Line 45: change these two sentences: “The virus could not be detected the in semen of men acutely infected with SARS-CoV-2 or recovered from COVID-19, although semen parameters are affected, so orchitis may be a complication following COVID-19.” **Be careful with the use of “COVID-19”. One has COVID-19 but they are infected with SARS-CoV-2.

Line 56: remove “infected”

Need to add something about whether COVID-19 causes complications in neonates, either citing from the literature or stating that no one has published about this. More should be stated about how the women in the previous studies were tested—same type of RT-PCR tests on vaginal swabs or was something else done?

Lines 59-60: move information about author contributions from methods to a new paragraph at the end of the manuscript titled “Author Contributions”. If you want to keep something about data collection and entry in the methods, remove the author initials and write that two people gathered and two people validated.

Line 63: move “were recruited” to just after “Women” at the beginning of the sentence and delete “to the study”.

Line 65: delete “infection”

Line 67: be specific about inclusion criteria around mild to moderately severe disease, and about exclusion criteria around severely ill.

Line 72: remove hyphen in “vaginal-sampling”

Line 75: citing viral transport medium after the swab is confusing. If you mean that the swab was placed in viral transport medium after collection, then write that. If you mean that the swabs and medium both came from Biological Industries, then state that more clearly. If the swabs came from a different company, cite that company.

Lines 78-83: the references to laboratories is confusing. First “Microbiology”, then “COVID-19”, then “our”.

Line 83: throughout this paragraph and double-check throughout the manuscript, make sure it is a lower case o in “SARS-CoV-2”

Line 85: capitalize CA (best if you can add the city name as well)

Line 86: add comma after SARS-CoV-2

Line 87: add comma after USA), capitalize CA here too

Line 88: was the choice per sample? Could reword this as “method used for each sample was chosen for convenience”

Line 92: add percentage after 16

Line 93: delete “and 35- non-pregnant patients”

Line 93: the high level of hospitalization emphasizes the need for clear criteria for inclusion and exclusion in the study

Line 96: delete “infected” since that is understood as a criterion for the study

Line 98 and Table 1: This is the first mention of chronic diseases and they are listed in the table footnote. Collection of this data should be detailed in the methods, such as what type of medical history was collected, and how this was collected (medical chart? Asking them?). Same for the other clinical data that are presented in the table. Were all of the tests done on all the women even if not hospitalized? Were X-rays done on all the women? Was CRP measured as part of clinical care or as part of the study?

Line 115: this is the first mention of a “study period”. Describe clearly in the methods how long women were followed for this study.

Line 138: This sentence needs more detail. The two separate techniques were done for both swabs or one was done for each swab?

Line 139: change to “very low levels”

Line 140: compare the vaginal results to the range of values that are obtained with nasopharyngeal swabs—is it lower than all of them or just lower than the NP swab from this woman? More should be said about the low Ct values for the vaginal swab tests—what is the data that these are high enough to consider them positive?

Line 143: add a comma after “study”

Line 145 and later in this section: no need to bold any words.

Line 146: change to “from the nasopharyngeal swab collected from the same woman”. Possibly change this whole sentence to indicate that the results were just at the limit of detection for the assay, not just lower than the NP swab.

Line 150: add hyphen in “COVID-19”

Line 170: too much is made of the “4 hours after the diagnosis”; the women were not diagnosed at the same time after infection, so the only thing that can be said is the that the vaginal sampling happened near the time that they had  high levels of virus that were sampled by NP swab.

Line 175: It is unclear from the information provided how the second PCR analysis was done—was this using the same sample as the Liaison test or with the second sample? Also, the Ct value for the second test is higher than for the first test, which makes it difficult to use as evidence of this being a true positive test. If the authors are going to spend many lines of text and a figure showing how confident they are that this is a positive test, they need to add more detail about the tests as noted above.

Line 178: this isn’t really so much of a comparison of pregnant vs non-pregnant women for vaginal SARS-CoV-2, and more of a survey of women including pregnant women.

Lines 178-180: The sentence starting “By this comparison” should be deleted since only one woman had a positive result.

Line 183: delete “as opposed to other studies,”

Tables 1 and 2: it would be helpful to show the range of values for the continuous variables.

Figure 1: It appears that lines 202-210 should be part of the legend and not the text. The sentence in lines 202-204 should be reworded to first say which test is shown in each part, then state the genes detected in that assay. In line 206, “lowest is” should be changed to “lower”, and delete the word is at the end of the sentence. Lines 206-208 are written incorrectly since the orange SPC line is in C; commas should be used rather than dashes. It is not clear what is meant by “These are SARS-CoV-2 culture supernatant (provided by the kit manufacturer)”. The last sentence about the results (Out of four genes…) should be deleted because it is results rather than explanation of the figure. There is no specification of which vaginal swab was used for either of the tests in B or C. This should be indicated here and in the text.

References:

1, 15, 29. fix the author name

2, 3, 4, 6, 7, 12, 13, 21. fix capitalization of the journal name or group name (WHO). Also remove extra journal info for refs 3, 4, 22.

3, 10, 12, 13, 21, 22, 25, 28, 30. need journal information. If only epub, need to say “Epub before print” or something similar.

7, 10, 24. use correct journal abbreviation: J. Am. Med. Assoc. and BJOG (all caps)

14, 17, 28. need an author

17. need year of publication

Reviewer 3 Report

This study investigated the presence of SARS-CoV-2 in vaginal secretions of pregnant and non-pregnant women infected with COVID-19. It is meaningful that the authors analyzed relatively large number of cases consisted with pregnant and non-pregnant women, because we cannot completely role out vertical transmission during vaginal delivery.

In addition, it there has been many reports of SARS-CoV-2 in semen and we have to discuss possible risks of sexually transmission. However, I expect the authors multiple sampling during the course of the disease to study its clinical kinetics. In addition, the detailed clinical picture of a 60-year-old case with SARS-CoV-2 should be provided.

Author Response

Reviewer 3- general comment:

            We wish to thank the reviewer by the remarks. We have made significant changes throughout the manuscript to accommodate.

Comment

This study investigated the presence of SARS-CoV-2 in vaginal secretions of pregnant and non-pregnant women infected with COVID-19. It is meaningful that the authors analyzed relatively large number of cases consisted with pregnant and non-pregnant women, because we cannot completely role out vertical transmission during vaginal delivery.

In addition, it there has been many reports of SARS-CoV-2 in semen and we have to discuss possible risks of sexually transmission. However, I expect the authors multiple sampling during the course of the disease to study its clinical kinetics. In addition, the detailed clinical picture of a 60-year-old case with SARS-CoV-2 should be provided.

Response:

We appreciate the reviewer’s comments.

We agree that it would have been ideal to perform multiple sampling during the course of the disease to study its clinical kinetics. However, we did not wish to burden the patients with multiple vaginal sampling. Consequently, we aimed to perform vaginal testing early in the disease course due to theoretical higher levels of the virus.

We have added the required information about the patient:

Lines 297-307: Only one patient (1.9%) had positive results for SARS-CoV-2 in 2 separate samples of vaginal secretions. The patient was 60 years old with a significant medical history of chronic diseases: systemic lupus erythematosus, antiphospholipid syndrome, and hypertension. In her past she underwent total abdominal hysterectomy due to menorrhagia and uterine leiomyomas. Her regular medications included: Methotrexate, Warfarin, Aspirin, Levothyroxine, Belimumab, and Hydroxychloroquine. She was diagnosed with SARS-CoV-2 following complaints of symptoms of dry cough and weakness and she was hospitalized for supportive care. At admission and during her hospitalization she was hemodynamically stable, without worsening of her symptoms. She was mobile, without other complaints. Repeat physical examinations were without additional pathological findings and she was discharged 4 days following her admission in good general condition.